# Post-Operative Permanent Hypoparathyroidism and Preoperative Vitamin D Prophylaxis

**DOI:** 10.3390/jcm10030442

**Published:** 2021-01-24

**Authors:** Tara Kannan, Yasmin Foster, David J. Ho, Scott J. Gelzinnis, Michael Merakis, Katie Wynne, Zsolt J. Balogh, Cino Bendinelli

**Affiliations:** 1School of Medicine and Public Health, University of Newcastle, Newcastle, NSW 2308, Australia; tara.kannan@uon.edu.au (T.K.); yasmin.foster@uon.edu.au (Y.F.); davidjianyuan.ho@uon.edu.au (D.J.H.); scott.gelzinnis@uon.edu.au (S.J.G.); michael.merakis@uon.edu.au (M.M.); katiejane.wynne@health.nsw.gov.au (K.W.); Cino.Bendinelli@health.nsw.gov.au (C.B.); 2Department of Endocrinology, John Hunter Hospital, Newcastle, NSW 2305, Australia; 3Department of Surgery, John Hunter Hospital, Newcastle, NSW 2305, Australia

**Keywords:** permanent hypoparathyroidism, postoperative hypocalcaemia, thyroidectomy, preoperative vitamin D

## Abstract

Permanent hypoparathyroidism, a feared thyroidectomy complication, leads to significant patient morbidity, medical treatment, and monitoring. This study explores whether preoperative high-dose vitamin D loading decreases the incidence of permanent hypoparathyroidism. In a subgroup analysis, the study examines the predictive utility of day 1 parathyroid hormone (PTH) in permanent hypoparathyroidism. Patients (*n* = 150) were previously recruited in the VItamin D In Thyroidectomy (VIDIT) trial, a multicentre, randomised, double blind, placebo-controlled trial evaluating the role of 300,000 IU cholecalciferol administered orally a week before total thyroidectomy. Patients were contacted postoperatively beyond six months through a telephonic questionnaire. The primary outcome was permanent hypoparathyroidism, strictly defined as the need for activated vitamin D six months postoperatively. Out of 150 patients, 130 (86.7%) were contactable. Permanent hypoparathyroidism occurred in 11/130 (8.5%) patients, with a lower incidence of 5.3% (3/57) in the cholecalciferol group compared to 11% (8/73) in the placebo group; however, this was non-significant (*p* = 0.34). In a subgroup analysis, no relationship between day 1 PTH level and the incidence of permanent hypoparathyroidism was found (*p* ≥ 0.99). There was a lower rate of permanent hypoparathyroidism in the cholecalciferol group, which was not significant. The predictive utility of day 1 postoperative PTH levels may be limited to transient hypoparathyroidism.

## 1. Introduction

Total thyroidectomy is required to treat several conditions, including thyroid cancer, Graves’ disease, large symptomatic nodules, and multinodular goitre. In the United States, it is performed in 60 operations per 100,000 patients per year [1]. The close anatomical proximity and shared vascular supply of the parathyroid and thyroid glands can lead to parathyroid devascularisation and transient or permanent hypoparathyroidism [2,3]. This presents clinically as hypocalcaemia, which occurs in 23% of patients after thyroidectomy [4]. Hypocalcaemia can lead to muscle spasms, arrhythmias, paraesthesia, tetany, seizures, and death [5].

Vitamin D is an important therapeutic target due to its vital role in intestinal calcium absorption. Deficiency results in secondary hyperparathyroidism [6]. In these patients, damage to parathyroid glands during thyroidectomy may lead to marked hypocalcaemia [7]. In patients with sufficient vitamin D levels, hypoparathyroidism may lead to less severe manifestations [5].

If hypoparathyroidism persists beyond six months, it is considered permanent [8]. After total thyroidectomy, the reported incidence of permanent hypoparathyroidism ranges from 0.5 to 2% [9], with a more recent registry-based study reporting a prevalence of more than 5% [8]. In these patients, there is a 17-fold increase in the risk of chronic kidney disease due to reduced renal calcium reabsorption, leading to hypercalciuria and nephrocalcinosis [10]. Other complications include cataracts, basal ganglia calcification, anxiety, and depression [10,11].

Predicting permanent hypoparathyroidism through early identification of patients that require monitoring would allow early intervention to prevent severe hypoparathyroidism requiring in-patient care [12]. Day 1 PTH status is a reliable predictor of postoperative hypoparathyroidism with a sensitivity of 83.4% and a specificity of 100% [13]. Day 1 PTH < 10 pg/mL is an effective determinant; however, in vitamin D-deficient patients who have secondary hyperparathyroidism, postoperative PTH can be an unreliable predictor [14]. The utility of day 1 PTH in predicting transient hypoparathyroidism is well documented, although its role in permanent hypoparathyroidism is lesser known.

Hypoparathyroidism in the immediate postoperative period may not translate to permanent hypoparathyroidism. Only one randomised study on preoperative vitamin D to prevent hypoparathyroidism has followed patients beyond the immediate postoperative period [9], showing perioperative alfacalcidol supplementation to decrease the incidence of permanent hypoparathyroidism (*n* = 219; 0% vs. 4%; *p* = 0.04).

This current study included participants previously recruited in the VItamin D In Thyroidectomy (VIDIT) trial [12]. The VIDIT trial was a randomised, double blind, placebo-controlled trial (RCT) with 150 patients, which evaluated the role of preoperative high-dose cholecalciferol (300,000 IU) in preventing post-thyroidectomy hypoparathyroidism. It concluded that preoperative cholecalciferol did not reduce the incidence of postoperative hypocalcaemia within six months of surgery (29% vs. 38%; *p* = 0.23). Most patients in the original study were vitamin D replete. When stratified by day 1 PTH status (≥10 pg/mL or <10 pg/mL), improved clinical outcomes were noted in both subgroups if receiving cholecalciferol (95% CI 0.32–0.98; *p* = 0.04). The positive outcomes included reductions in hypocalcaemia, length of hospital stay, and requirement for postoperative therapy. The VIDIT trial provided an opportunity to assess the effect of vitamin D supplementation on outcomes beyond six months after operation. Permanent hypoparathyroidism has been defined as an activated vitamin D requirement six months after surgery, as per the recent paper from Almquist et al. (2018).

It was hypothesized that preoperative high-dose vitamin D supplementation would decrease the incidence of permanent hypoparathyroidism.

## 2. Methods

The VIDIT was a multicentre RCT aiming to demonstrate the role of preoperative vitamin D loading in preventing postoperative hypocalcaemia and hypoparathyroidism. The study was approved by the Hunter New England Human Research Ethics Committee (AU202004-02). Patients were eligible for the trial if they underwent total or completion thyroidectomy. Recruitment occurred in the John Hunter Hospital and Newcastle Private Hospital from August 2014 to December 2017. Eligible and consenting patients were allocated using centralised randomisation to either the vitamin D (single oral dose of 300,000 IU of cholecalciferol) or placebo group. All thyroidectomies were performed by four high-volume thyroid surgeons. The details of the eligibility criteria and treatment protocol have been published previously [12].

The primary outcome measured in this study was the incidence of permanent hypoparathyroidism, defined as the need to use activated vitamin D six months after surgery. The subgroup analysis evaluated the predictive value of low day 1 PTH, defined as day 1 PTH < 10 pg/mL. Data was collected through a telephonic questionnaire where patients were asked about current calcitriol use (Appendix A
Figure A1). If the patient was uncontactable after 3 attempts, the patient’s general practitioner was contacted. Day 1 PTH levels were previously collected in the VIDIT trial.

Variables were tested for normality via the D’Agostino and Pearson normality test. All variables reached significance, failing the test, and therefore were treated as non-parametric data. Continuous variables such as age and body mass index (BMI) were tested using an unpaired Mann–Whitney U test, and our dichotomous categorical variables were tested using a Fisher’s exact test for independence. Statistical analysis was performed using GraphPad Prism version 7.00 for Windows (La Jolla, CA, USA).

The cohort was not adequately powered to investigate the primary outcome of permanent hypoparathyroidism. The original VIDIT trial was powered at 80% with classification of the endpoint using a variety of measurements, such as prolonged QT interval on ECG associated with biochemical markers and symptomatic presentations of hypocalcaemia in a hospital setting. Due to the limited, simplistic nature of the classifying dichotomous endpoint and significant loss to follow-up, our post-hoc power calculation was estimated as 20.1%. The significant decrease in follow-up sample size increased the threshold of a potential type II error to 80%, far more than the traditionally acceptable 20%. Due to the unacceptably high risk of type II error, the ability to draw statistical conclusions is limited.

## 3. Results

### 3.1. Participant Flow

The VIDIT trial analysed the data of 150 patients who underwent thyroidectomy. Of these, 130 patients were potential participants (Figure 1). However, 18.6% of patients from the cholecalciferol group and 8.6% of patients from the placebo group were unavailable for analysis. This resulted in a total loss to follow-up of 13.3% (*n* = 20).

### 3.2. Baseline Characteristics

Table 1 presents the baseline characteristics of study patients available for follow-up. Overall, mean age was 54.5 years, and 76.7% were female. Of the operations, 46.9% were for goitre, 30% were for Graves’ disease, and 23.1% were for thyroid cancer. Mean pre-enrolment 25-OH vitamin D was 72.9 nmol/L.

### 3.3. Complete Case Analysis

To ensure that the randomisation from the original RCT was preserved and that a complete case analysis approach did not introduce bias into the data set, statistical analysis was carried out on the baseline characteristics between the group lost to follow-up and the group included for analysis. Due to the small sample size (*n* = 20) and the larger (*n* = 130) group not passing normality testing, continuous variables were analysed using a Mann–Whitney U test. Categorical variables were similarly treated with a non-parametric Fisher’s exact test. Neither post-hoc analysis of effect sizes nor predictive analysis such a regression were carried out, due to lack of significant findings.

Age, sex, BMI, length of stay, surgical indication, and pre-dose vitamin D were minimally variable (Table 2). In all metrics, *p*-values did not reach significance. We can conclude that the decision to exclude patients without any data for their primary outcome did not introduce bias or dilute the effect of randomisation.

### 3.4. Loss to Follow-Up

Univariate analysis on the group that was included in the analysis and the group that was lost to follow-up was unable to disprove the assumption that the patients lost to follow-up were at random (*p* > 0.05), therefore we accept that the effect of randomisation is preserved. To prevent loss to follow-up, 11/130 (8.5%) general practitioners were contacted as an alternative when their patients could not be contacted after three attempts (Figure 1).

### 3.5. Primary Outcome

Permanent hypoparathyroidism, defined as an activated vitamin D requirement six months after surgery, occurred in 3/57 (5.3%) participants taking cholecalciferol and 8/73 (11%) participants taking placebo (*p* = 0.34). Of these patients (*n* = 11), 6/11 (54.5%) suffered from hypocalcaemia, defined as corrected serum calcium < 2.10 mmol/L within six postoperative months. The calcitriol dosage ranged from 0.25 mcg to 1 mcg daily. There were no statistically significant differences in dosage between the intervention (*n* = 3) and placebo (*n* = 8). Dosage was not associated with other factors such as surgical indication, sex, age, or BMI. This dichotomous, categorical primary outcome was assessed using a Fisher’s exact test for independence, which showed no significant, independent effect (*p* = 0.34) between preoperative supplementation with high-dose cholecalciferol and the incidence of post-thyroidectomy permanent hypoparathyroidism.

### 3.6. Secondary Findings—Sub-Group Analysis

The original trial noted divergent rates of postoperative hypocalcaemia when stratified by day 1 postoperative PTH status. This was explored in this cohort for both the incidence of day 1 hypoparathyroidism between study groups and the presence of an independent effect between low postoperative PTH and permanent hypoparathyroidism. Day 1 hypoparathyroidism was defined as a PTH level < 10 pg/mL. The incidence in the intervention vs. placebo group was 24.6% and 30.1%, respectively. A Fisher’s exact test for independence failed to display a significant difference between groups (*p* = 0.56).

Subgroup analysis was then carried out, first stratifying the cohort based on day 1 postoperative PTH levels. Patients were dichotomised with either a PTH reading of <10 pg/mL or ≥10 pg/mL. An independent relationship between day 1 postoperative PTH level and the incidence of permanent hypoparathyroidism was explored with a Fisher’s exact test. Almost identical rates of permanent hypoparathyroidism (8.3%, 8.5%) were observed between the two groups, and the Fisher’s exact test failed to reach significance (*p* = > 0.99) (Table 3).

## 4. Discussion

Permanent hypoparathyroidism can be associated with long-term hypocalcaemia symptoms (if not medicated) and ectopic calcification [10,11]. This study followed patients in the VIDIT trial to ascertain whether preoperative cholecalciferol loading reduced the rate of post-thyroidectomy permanent hypoparathyroidism [12]. This research demonstrated that permanent hypoparathyroidism occurred at a numerically lower rate in the cholecalciferol group (5.3%) compared to the placebo group (11%); however, this result was not statistically significant. This was consistent with the finding of the original VIDIT trial which showed that preoperative cholecalciferol did not reduce the incidence of postoperative hypoparathyroidism within six months of surgery (29% vs. 38%; *p* = 0.23). This study also analysed day 1 postoperative PTH levels, which are a useful marker of postoperative hypoparathyroidism. Rowe et al. (2019) conducted a subgroup analysis dichotomized by day 1 PTH status (<10 pg/mL, below the lower limit of the local reference range; ≥10 pg/mL, normal); they found more favourable clinical outcomes in the participants of both the low and normal PTH subgroups who had received preoperative cholecalciferol (95% CI 0.32–0.98; *p* = 0.04). This current study found no significant between-group difference in the incidence of permanent hypoparathyroidism (8.3% vs. 8.5%; *p* ≥ 0.99), suggesting that day 1 PTH < 10 pg/mL may only be predictive of transient hypoparathyroidism. While day 1 PTH has a sensitivity of 83.4% and a specificity of 100% for predicting postoperative hypoparathyroidism, its utility may be limited in vitamin D-deficient patients who have secondary hyperparathyroidism [13,14].

As discussed earlier, vitamin D is an important therapeutic target as deficiency leads to secondary hyperparathyroidism [6]. Intraoperative damage to the parathyroid glands can lead to marked hypocalcaemia in these patients [7]. It is postulated that vitamin D < 37.4 nmol/L is a risk factor for postoperative hypoparathyroidism [5]. However, since these patients were largely vitamin D replete, the effect size could have been limited in this cohort.

The overall incidence of permanent hypoparathyroidism was higher than anticipated at 8.5% compared to the expected value of 0.5–5.4% [8,9]. This higher rate is likely due to a small sample size and loss to follow-up, where those patients who were successfully followed-up may be more health conscious to begin with, and hence found to have permanent hypoparathyroidism and require calcitriol, as opposed to those lost to follow-up. However, the reported rates of permanent hypoparathyroidism vary widely from 0.2 to 10% due to differing definitions [15,16,17,18].

The definition of permanent hypoparathyroidism varies in the literature and often comprises three primary domains: a quantitative domain based on PTH level, a qualitative domain based on clinical features or activated vitamin D requirement, and a temporal domain based on time since operation [19,20,21,22,23,24]. This study found that 15 patients used calcium carbonate, of which eight used it alone, and hence, did not meet the definition for permanent hypoparathyroidism. Calcium carbonate was not included in the definition since it is not a specific treatment option and is more often used for alternative indications, such as osteoporosis and general bone health. One young patient was also found to use recombinant PTH supplementation. Unlike other hormonal insufficiencies, permanent hypoparathyroidism is not treated through replacement therapy with PTH in Australia, although it has recently been approved in the United States for refractory cases [25].

This study builds on results from one previous randomised trial of perioperative activated vitamin D supplementation. Genser et al. (2014) found that the incidence of permanent hypoparathyroidism was reduced in a group receiving alfacalcidol compared to a placebo group (0% vs. 4%, *p* = *0*.04).

There are several strengths of this trial. First, it follows an RCT and therefore has decreased selection bias, and between-group similarity of baseline characteristics. Second, the involvement of the patient’s general practitioner for data collection if the patient was not contactable decreased loss to follow-up. Finally, this study is unique in evaluating the long-term effectiveness of a single high preoperative dose of cholecalciferol, which is considered superior to active vitamin D due to its low toxicity [12,26].

A limitation is that the definition used for permanent hypoparathyroidism only considered activated vitamin D supplementation, and hence failed to consider clinical features, biochemistry results, and calcium supplementation. This was deemed a valid definition in a recent paper by Almquist et al. (2018), since hypocalcaemia can occur in the setting of normal PTH levels. Activated vitamin D requirement is also more specific for capturing patients with permanent hypoparathyroidism than calcium requirement, and its use is monitored by a specialist, minimizing unnecessary use. Given the stringent diagnostic criteria, measuring the primary outcome is precise and reproducible, and any interobserver variability is minimised. Additionally, the robustness of the follow-up method was undermined by recall bias, failed contact attempts, contacting patients at differing postoperative time points, and limiting follow-up to a single phone call.

In conclusion, a lower rate of permanent hypoparathyroidism was observed in the group taking high-dose preoperative cholecalciferol, however this was not statistically significant. Additionally, despite day 1 PTH being a reliable predictor in transient hypoparathyroidism, it did not predict permanent hypoparathyroidism. Further research with larger sample sizes, more regular postoperative follow-up, and a definition inclusive of biochemistry results is required to understand the interaction between preoperative vitamin D loading and the incidence of permanent hypoparathyroidism. Reducing the risk of permanent hypoparathyroidism can not only reduce patient morbidity but can also negate the need for long-term activated vitamin D treatment, routine biochemical monitoring, and regular symptom assessment.

## Figures and Tables

**Figure 1 jcm-10-00442-f001:**
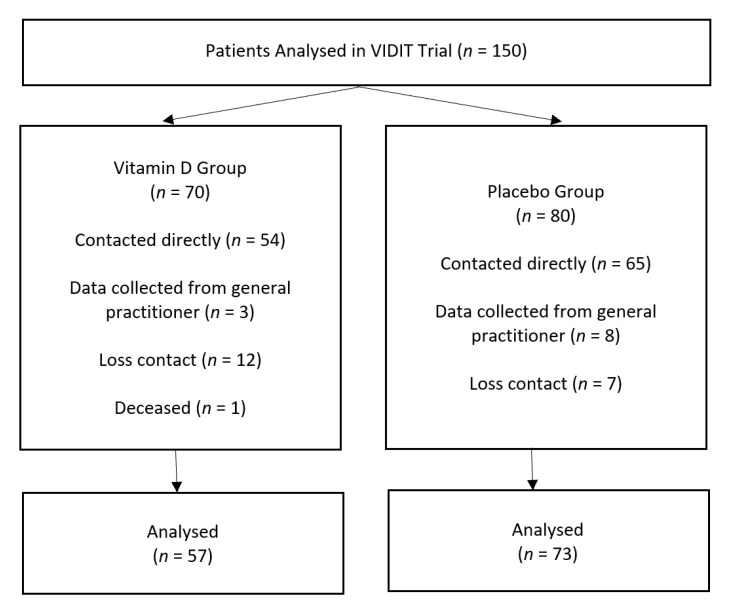
Patients excluded for analysis from original VIDIT trial.

**Table 1 jcm-10-00442-t001:** Demographics and clinical parameters.

*n* = 130	Intervention (*n* = 57)	Placebo (*n* = 73)
Age: median (Q1, Q3)	53 (34.5, 61.5)	56 (40.5, 66.5)
Female: *n* (%)	43 (75.4%)	57 (78%)
Body mass index: median (Q1, Q3)	28.3 (24.9, 32.3)	30.0 (26.3, 34.9)
Length of hospital stay: median (Q1, Q3)	2 (2, 2)	2 (2, 3)
Graves’ Disease: *n* (%)	18 (31.6%)	21 (28.8%)
Goitre: *n* (%)	24 (42.1%)	37 (50.7%)
Thyroid cancer: *n* (%)	15 (26.3%)	15 (20.5%)
Pre-dose 25-OH vitamin D (nmol/L) median (Q1, Q3)	73.2 (58, 90)	72.5 (58, 90)

**Table 2 jcm-10-00442-t002:** Baseline characteristics: lost to follow-up vs. included in analysis.

	Lost Contact	Analysed	*p*-Value
*n* = 150	20	130	
Age: median (Q1, Q3)	53 (45.5, 60.2)	55 (38, 63.3)	0.78
Female: (%)	14 (70%)	100 (78%)	
Body mass index: median (Q1, Q3)	28.1 (24.7, 34.1)	29.3 (25.5, 33.7)	0.75
Length of hospital stay: median (Q1, Q3)	2 (2, 2.75)	2 (2, 2)	0.68
Graves’ Disease: *n* (%)	7 (35%)	39 (30%)	0.70
Goitre: *n* (%)	10 (50%)	61 (46.9%)	
Thyroid cancer: *n* (%)	3 (15%)	30 (23.1%)	
Pre-dose 25-OH vitamin D (nmol/L): median (Q1, Q3)	72 (54.25, 79.75)	72 (57.75, 90)	0.4

**Table 3 jcm-10-00442-t003:** Incidence of permanent hypoparathyroidism: day 1 PTH ≤ 10 pg/mL vs. day 1 PTH > 10 pg/mL.

	*n* = 130	Day 1 PTH < 10	Day 1 PTH ≥ 10	*p*-Value
*n*		36	94	
Permanent Hypoparathyroidism	Yes = 11 (8.4%) No = 119 (91.6%)	3 (8.3%) 33 (91.7%)	8 (8.5%) 86 (91.5%)	>0.99

## Data Availability

The data presented in this study are available on request from the corresponding author. The data are not publicly available due to privacy and ethical reasons.

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
