# Peer review of "Post-Operative Permanent Hypoparathyroidism and Preoperative Vitamin D Prophylaxis"

_jcm, 2021, doi:10.3390/jcm10030442_

Round 1

Reviewer 1 Report

Line 34 please precise 60 per 100000 patients

38 can lead

42 can lead or may lead

52 How? Eplain!

169 can be

170 ectopic calcification is extrem rare, please add literature

180 significance

195 follow-up

203 please use generic name

219 hypothetical, add literature

The limitation of the study is the definition of hypoparathyroidsm based on the supplementation of activated vitamine D alone. To define hypoparathyroidism at least calcium and PTH levels 6 months postoperatively are necessary (if not even vitamine D levels). There are  patients whith hypoparathyroidism who are under calcium supplementation and there may be also patients who take vitamine d without need.

Therefore calcium and PTH levels after 6 months in patients who presented hypoparathyroidsm postoperatively are necessary. 

Reviewer 2 Report

No changes, great work

Reviewer 3 Report

The data analysis  is appropriate, methods are adequately described and the conclusions are supported by the results. 

In line 211 there is a repat error (that that)

English language and style are adeguate

The authors are critical of both strengths and limitations of the paper

English language and style are adeguate
